# Wooden and Plastic Pallets: A Review of Life Cycle Assessment (LCA) Studies

**Ivan Deviatkin [1],\*** , **Musharof Khan [1], Elizabeth Ernst [2] and Mika Horttanainen [1]**

1   Department of Sustainability Science, School of Energy Systems, Lappeenranta-Lahti University of Technology LUT, 53850 Lappeenranta, Finland; musharof.khan@lut.fi (M.K.); mika.horttanainen@lut.fi (M.H.)
2   Future Sustainable foods Research Group, Department of Agriculture and Forestry, Helsinki University, 00100 Helsinki, Finland; elizabeth.ernst@helsinki.fi
\*   Correspondence: ivan.deviatkin@lut.fi

**Abstract:** Pallets are the tiny cogs in the machine that drive transportation in the global economy. The profusion of pallets in today's supply chain warrants the investigation and discussion of their respective environmental impacts. This paper reviews the life cycle assessment studies analyzing the environmental impacts of pallets with the intent of providing insights into the methodological choices made, as well as compiling the inventory data from the studies reviewed. The study is a meta-analysis of eleven scientific articles, two conference articles, two peer-reviewed reports, and one thesis. The review was implemented to identify the key methodological choices made in those studies, such as their goals, functional units, system boundaries, inventory data, life cycle impact assessment (LCIA) procedures, and results. The 16 studies reviewed cumulatively analyzed 43 pallets. Mostly pooled ($n = 22/43$), block-type ($n = 13/43$), and wooden ($n = 32/43$) pallets with dimensions of 1219 mm × 1016 mm or 48 in. × 40 in. ($n = 15/43$) were studied. Most of the studies represented pallet markets in the United States ($n = 9/16$). Load-based (e.g., 1000 kg of products delivered), trip-based (e.g., 1000 trips), and pallet-based (e.g., one pallet) functional units were declared. A trip-based functional unit seems the most appropriate for accounting of the function of the pallets, as its purpose is to carry goods and facilitate the transportation of cargo. A significant amount of primary inventory data on the production and repair of wooden and plastic pallets are available, yet there are significant variations in the data. Data on pallets made of wood–polymer composites was largely missing.

**Keywords:** life cycle assessment (LCA); carbon footprint; life cycle inventory; literature review; wooden pallet; plastic pallet; composite pallet

## 1. Introduction

In the existing world of increasing mobility and growing trade, material and commodities need to be transported safely across various actors of the economy: from suppliers to manufacturers, from manufacturers to warehouses, from warehouses to retailers, and finally to consumers. As global trade increases, the role pallets' play becomes ever more significant, because they literally move the world. According to the market estimated by Freedonia [1], the global demand on pallets was expected to slightly surpass 5 billion pallets in 2017, out of which roughly 30% would be supplied in North America, 20% in Western Europe, and 30% in the Asia Pacific.

Pallets, as defined in the SFS-EN ISO 445 standard [2], represent "rigid horizontal platforms of minimum height, compatible with handling by pallet trucks and/or forklift trucks and other appropriate handling equipment, used as a base for assembling, loading, storing, handling, stacking,

transporting, or displaying goods and loads". Pallets can be made of various materials (softwood, hardwood, plastic, cardboard, aluminum, and composites), in various forms (stringer, block, reversible, two-way, four-way, nestable, etc.), and of various dimensions (European (EUR) size: 1200 mm × 800 mm, Finnish (FIN) size: 1200 mm × 1000 mm, Grocery Manufacturers Association (GMA) size: 48 in./1219 mm × 40 in./1016 mm, etc.).

Pallets, despite being simple in design, often undergo vastly different lifecycles depending on management strategy [3,4]. There are three pallet management strategies that dominate the industry: single-use, buy/sell, and pooled. Single-use is the simplest strategy wherein pallets are discarded after one trip. However, standardized pallets are usually designed to last for several trips. Such pallets can be either operated using so-called "buy/sell" strategy, where ownership of pallets is transferred together with the pallets or alternatively, the pallets can be managed using a "pooling" strategy where pallets are leased to customers without transfer of the ownership. In the pooling strategy, pallets are usually marked in a company-specific way (e.g., by using a specific color, making it possible to track the lifecycle of the pallets). Furthermore, radio frequency identification (RFID) trackers are increasingly used by pallet poolers, which allows for data collection on the pallet's lifecycle and location throughout the supply chain [5]. In the buy/sell strategy, pallets are freely exchanged on the market making it impossible to precisely estimate their use intensity. Pallets can also be repaired during their lifecycle to prolong their service life [6].

The increasing demand on pallets, the growing competition on the market, as well as the introduction of novel materials, such as composites, have mutually driven the need to assess their environmental impacts. A significant body of research has accumulated which was partly embodied into the product category rules developed in North America [7]. However, no comprehensive review of the literature available has been published until now.

## 2. Review Process and Studies Reviewed

### 2.1. Review Process

This study systematically reviews literature on the life cycle assessment of pallets as outlined in Figure 1. The review was performed using the search strings of "pallet*" and "life cycle assessment" in title, abstract, and keywords of SCOPUS [8] and Web of Science [v.5.33] [9] databases. The search yielded 21 results for each database (42 in total). Apart from the sources identified in the databases, six other sources were found on the Internet. A total of 34 studies were left after the removal of duplicates. Out of them, four studies could not be retrieved by the authors, namely [10–13]. Thus, 30 articles were assessed for their eligibility to be included in the review process, out of where 14 studies were outside the scope of this paper. The studies excluded did not perform a life cycle assessment (LCA) of pallets, but rather included pallets as a part of their system boundaries. A total of 16 studies were left for qualitative and quantitative analysis.

### 2.2. Studies Reviewed

In total, 16 studies were included in the current review (Table 1). The majority of the included studies (*n* = 11) were scientific journal articles. Most of the studies reviewed focused on providing life cycle inventory (LCI) data on pallet production, manufacturing, as well as other life cycle stages. Other studies focused on the comparison of pallets produced from different materials, such as wood and plastics. A study by Alvarez and Rubio [14] examined the use of different carbon footprint accounting methods using pallets as a case study. A study by Alanya-Rosenbaum et al. [15] was methodological in nature neither providing LCI nor comparing different pallets.

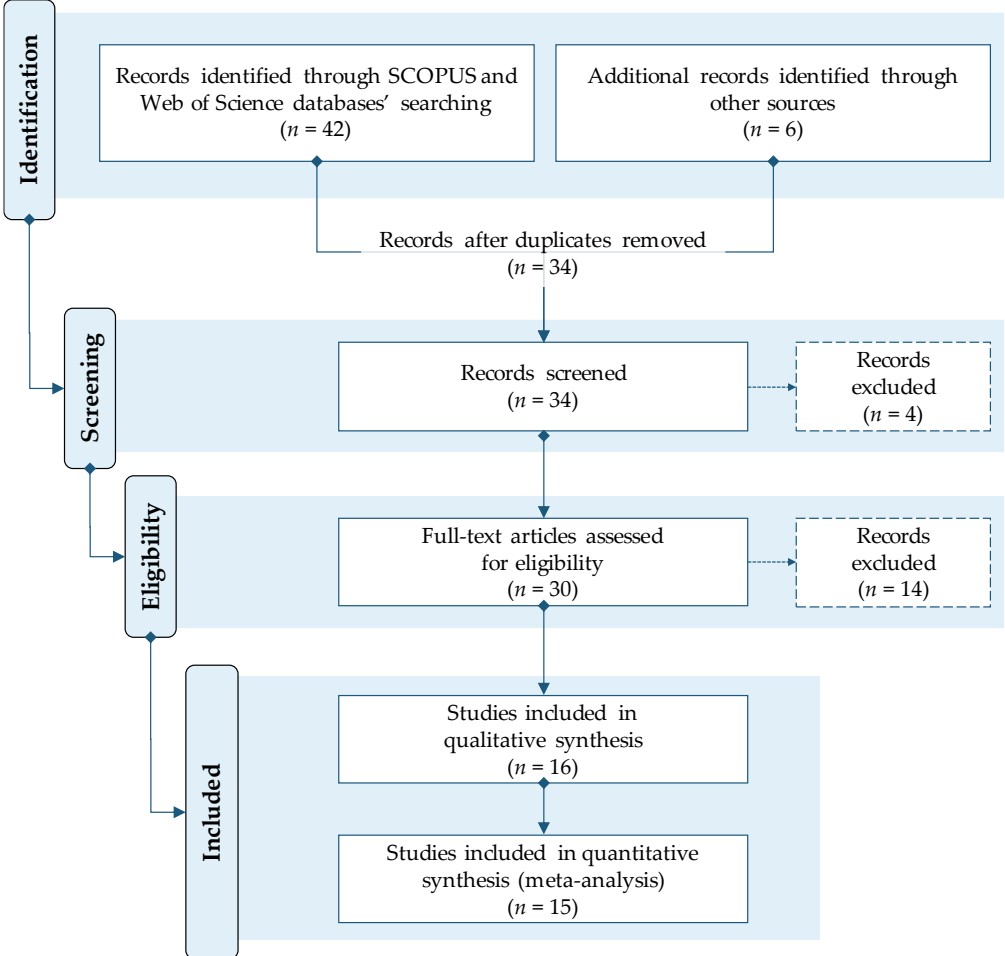

**Figure 1.** A flow diagram of the literature review implemented after Moher et al. [16].

## 3. Pallets Classification

Pallets can be classified based on their raw materials, type, dimensions, and management strategy (Figure 2). Considering raw materials, most of the pallets are made of wood (softwood, hardwood, technical wood), plastic (virgin plastic, recycled plastic), cardboard, or wood–polymer composites. The European market is dominated by the EUR pallet, size 1200 mm × 800 mm, whereas the markets of North America are mainly occupied with GMA-sized pallets with dimensions of 48 in./1219 mm × 40 in./1016 mm. Markets in China or Finland commonly have pallets with dimensions of 1200 mm × 1000 mm, whereas Australian pallets oftentimes have rectangular dimensions (1165 mm × 1165 mm). Pallets of other dimensions also exist and can be customized to meet customer demand. Currently there is no internationally recognized pallet standard, but the ISO sanctions six pallet types ranging in size from 800 × 1200 mm to 1016 × 2019 mm [17]. Pallets are commonly structured in one of two ways: either as a stringer or block pallet. The former is most commonly used in North America, whereas the latter is widely used in Europe. Both pallet types have equal bearing load capacities. As discussed in the introduction, pallets can also be classified by their pallet management strategy. Pallets intended for single use are commonly called "single-use pallets", while those are also called one-way, expendable, or non-pooled pallets. Pallets, whose ownership is transferred with the pallet are commonly known as "buy/sell pallets", which are also known as open-loop or exchange pallets. "Pooled pallets", also referred to as closed-loop, leased, cross-docking, or take-back pallets, are rented from a pallet management company who externally manages the pallet pool. It is worth mentioning that this classification is not strict with possible inconsistencies (e.g., buy/sell pallets are also "non-pooled" pallets, whereas cross-docking and take-back are specific methods of pallet pooling).



However, such classification attempts at harmonizing the terminology used across literature in different countries and continents.

**Table 1.** Studies reviewed, their type, and the main goal.

| No. | Reference | Publication Type [1] | The Main Goal of the Study |
|-----|-----------|----------------------|----------------------------|
| (1) | [18] | R | To provide a life cycle inventory (LCI) that quantifies resource and energy use, waste, and emissions associated with three pallet systems with different reuse, repair, and recycling rates. |
| (2) | [19] | J | To develop a life cycle inventory analysis and to analyze the environmental impacts of the current management system by means of a life cycle inventory assessment. |
| (3) | [20] | T | To address a void in the studies by presenting an unbiased comparative life cycle analysis (LCA) study comparing plastic and wooden pallets through investigation of their environmental impacts and carbon footprints. |
| (4) | [21] | J | To frame and model the environmental issues and impacts associated with the management of pallets throughout the entire life cycle. |
| (5) | [22] | J | To develop a parametric model describing LCI of a range of wooden pallets used as tertiary packaging. |
| (6) | [23] | J | To help increase the understanding of the impacts of decisions at each life-cycle phase of pallets and, by extension, returnable containers and other forms of packaging. |
| (7) | [24] | J | To assess, quantify, and compare the carbon emissions of recycled wood waste (technical wood) with virgin softwood in the application of wooden pallets using comparative carbon footprint assessment methodology. |
| (8) | [14] | J | To assess the potential of the compound method based on accounting for product carbon footprints. To evaluate the differences between this method and product carbon footprint. |
| (9) | [25] | C | To calculate the environmental impact of softwood (structural grade pine), hardwood, and plastic pallets compared to their key market alternatives: simple/one-way pallets of softwood or cardboard. |
| (10) | [26] | J | To provide a detailed comparison of the environmental impacts of the three pallet management strategies (single-use expendable, reusable buy/sell, and reusable leased pool) in each of the phases of the pallet lifecycle. |
| (11) | [27] | J | To explore and quantify the carbon equivalent ($CO_2$ eq.) emissions associated with the remanufacturing operations of wood pallets while considering loading and service environment conditions. |
| (12) | [28] | R | To undertake a comparative study of the environmental credentials of various pallet options using life cycle assessment (LCA). |
| (13) | [15] | C | To provide guidance to the wooden pallet sector for environmental performance assessment and to enhance knowledge for developing environmental product declarations (EPDs) in the wood pallet manufacturing industry. |
| (14) | [6] | J | To develop an LCI for the repair process of 48 × 40 in. (1219 × 1016 mm) stringer-class in the United States. |
| (15) | [29] | J | To determine the maximum distance at which a repair facility can be located so that a closed-loop pallet system is both environmentally and economically sustainable. |
| (16) | [30] | J | To build a life cycle model for both wood and plastic pallets. |

[1] Report (R); journal article (J); thesis (T); and conference proceedings (C).

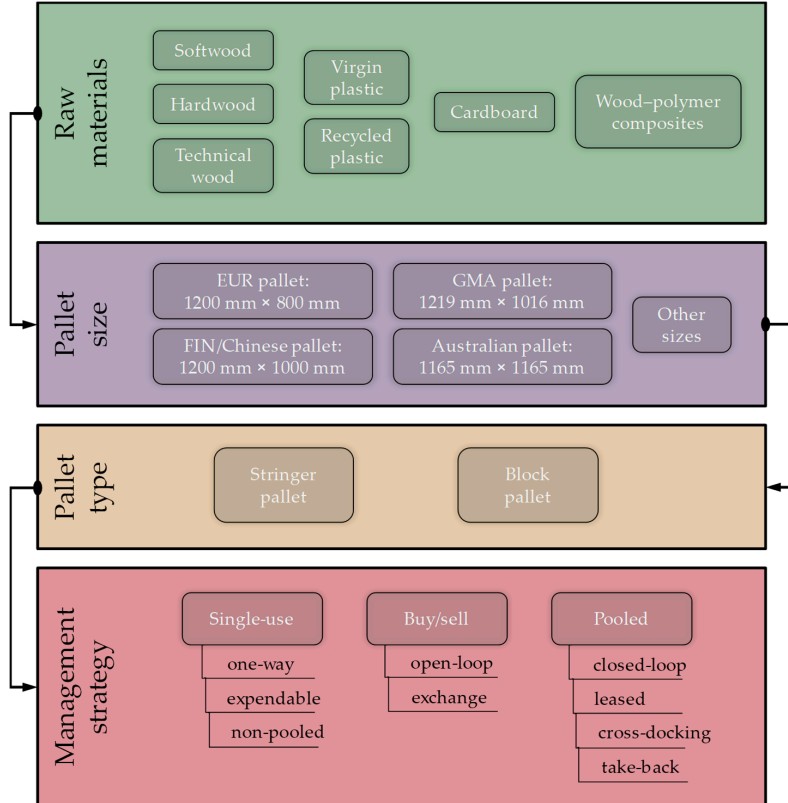

**Figure 2.** Classification of pallets based on their raw materials, size, type, and management strategy.

## 4. Methodological Choices

The studies reviewed employed a range of different methodological choices which are summarized in Figure 3. The majority of studies were comparative in nature. The comparisons made were between the raw materials used, which were most commonly wood and plastic, the different management strategies, as well as between differing usage intensities. The studies were often performed following the guidelines expressed in ISO 14040 standard [31] on LCA, whereas some of them followed the standards or guidelines on carbon footprints, such as ISO 14067 [32]. The reports included were critically reviewed ensuring their compliance with the ISO 14040 standard.

### 4.1. Geographical Scope

Geographical coverage of the reviewed studies varied, while the majority of the studies were representative of the situation in the United States or North America. Particularly, researchers from Auburn University, Rochester Institute of Technology, Virginia Tech, and the National Wooden Pallet and Container Association were actively involved in the research. Gasol et al. [19] and Alvarez and Rubio [14] studied pallet production in Spain, Kočí [30] assessed the impacts of pallet production in the Czech Republic, whereas Niero et al. [22] conducted a more generic study representative of European conditions. Finally, Bengtsson and Logie [25] and researchers from Edge Environment Pty Ltd [28] conducted their studies for pallet production in Australia and Southeast Asia. Such variation in the geographical representativeness might affect the results of the studies to the extent of variation in the LCI datasets of electricity generation, production of raw materials, as well as the impact of transportation and diesel production.

### 4.2. Functional Unit

Across the studies, a large variation was revealed in the functional unit chosen. Some studies set the number of trips as the functional unit (e.g., 100,000 trips [18,26] or 1000 trips [15,25]). Such

functional unit allows for a more comprehensive assessment of the different types of pallets having a different number of cycles per service life. It also accounts for the prolonged service life during pallet repair. The term cycle, in this context, refers to one trip through a supply chain from the highest echelon (the supplier, producer, or manufacturer) to the lowest echelon (the customer or end user). Though it should be noted that supply chains vary in distance and complexity so it is difficult to gauge exactly how many trips or cycles a pallet will last. In some studies [19,30], the mass of products delivered (e.g., 1000 t or 1000 kg), was chosen as a functional unit. This approach accounts for the differing carrying capacities of the pallets. It should be acknowledged that pallets cannot always support their specified maximum load. Deformation during production, high repair intensity, or damaged wood may result in weaknesses that lower the overall carrying capacity. Several studies [6,14,22–24] performed their research using a single pallet as the functional unit. In this case, studies should clearly indicate the carrying capacity of the pallets, as well as their expected number of cycles. This is needed to be able to convert the results to the number of trips since the primary function of the pallets is to transport goods. The number of pallets, on the contrary, could be recommended as a reference flow. The difference in the functional units complicates cross-comparison of studies and their effective discussion.

### 4.3. System Boundary

All stages of the pallets' lifecycle were studied in most of the articles reviewed, while their inclusion was limited in some of the studies. The cradle-to-grave studies typically included the impacts from the supply of its raw materials, such as wood, plastic, nails, electricity, and heat, the assembly of the pallets, their distribution and use, which mainly was the transport of the pallets to the customer and back, optional maintenance and repair, and finally recycling or disposal. Some of the studies [26–29] also studied the impact of the various handling and loading conditions on the results, which occur through altering the need for repair and changes in the number of cycles in their service life. In two studies [6,27] the impact of pallet manufacturing was left outside the system boundaries due to the specific impact of those studies on the repair strategies of the pallets.

### 4.4. Life Cycle Inventory

In many studies [6,14,18,19,22,28,30], primary and site-specific data were used to model the process of pallet manufacturing, as well as data on the consumption of raw materials and energy. In some other studies, combinations of primary and secondary data were used. However, not all studies were sufficiently transparent to ensure replication. Those that were transparent enough were further analyzed and the data was retrieved into a tabular form as presented in Section 6. Most of the modeling was performed in SimaPro, whereas many studies did not mention the software used. One study used Gabi for modeling.

### 4.5. Life Cycle Impact Assessment

Global warming potential, also referred to as carbon footprint or impact on climate change, was the most commonly studied impact assessment category. The impact was mostly modeled using CML 2000, ReCiPe (H), and TRACI impact assessment methods. However, the impact assessment methods were not clearly stated in many studies. Other non-toxic impact categories, such as acidification, eutrophication, or ozone layer depletion potential, were also studied [19,20,22,28,30]. The studies by Kurisunkal [20], Bengtsson and Logie [25], and Koči [30] assessed a wide spectrum of impacts, including non-toxic impact categories, toxic impacts, resource depletion potential, water use, as well as land occupation. In the study by researchers from Edge Environment Pty Ltd [28], emissions of biogenic carbon due to deforestation were accounted for, as well as an indicator accounting for the use of waste in the plastic pallets was calculated. Apart from the impact assessment, inventory data can be used to calculate indicators of circularity for products utilizing waste. One example of such an indicator could be the material circularity indicator developed by the Ellen MacArthur Foundation [33].

| No. | 1.1 Goal Definition | | 1.2 System Definition | | | 2. Life Cycle Inventory | | | 3. Life Cycle Impact Assessment | | | | | | |
| --- | Compa-rative nature | Key metho-dology | Functional unit | System boun-dary [a] | Geogra-phical scope [b] | Data for foreground system | Trans-parent inventory [c] | Software used | Method | GWP+ others [d] | Toxic | Reso urce | Water | Land |
| (1) | ■ | ISO 14040 ISO 14025 | 100,000 trips | ■■□■ | US, CA | Primary | □ | ? | ? | ■ | □ | □ | □ | □ |
| (2) | ■ | ISO 14040 | 1000 t products delivered | ■■□■ | ES | Primary | ■ | SimaPro | CML 2000 | ■■ | ■ | ■ | □ | □ |
| (3) | ■ | ? | Not specified quantitatively | ■[f]■□■ | US | Secondary | ■ | SimaPro | CML 2000, Impact 2002+, Eco-indicator 99 | ■■ | ■ | ■ | ■ | ■ |
| (4) | ■ | ? | ? | ■■□■ | US | Secondary | □ | ? | ? | ■ | □ | □ | □ | □ |
| (5) | ■ | ISO 14040 | 1 pallet | ■■□■ | RER [b1] | Primary | ■ | ? | ReCiPe (H) | ■■ | ■ | ■ | | ■ |
| (6) | ■ | ? | 1 pallet | ■■■■ | US | Mixed | ■ | SimaPro | IPCC | ■ | □ | □ | □ | □ |
| (7) | ■ | ISO 14040, BSI 2008 | 1 pallet (system) | ■■□■ | ? | Secondary | ■ | ? | ? | ■ | □ | □ | □ | □ |
| (8) | ■ | ISO 14067 | 1 pallet | ■■□□ | ES | Primary | ■ | ? | ? | ■ | □ | □ | □ | ■ |
| (9) | ■ | ISO 14040 | 1000 trips | ■■□■ | AU, CN | Mixed | ■ | SimaPro | ReCiPe (H) | ■ | ■ | ■ | ■ | ■ |
| (10) | ■ | ? | 100 000 trips | ■■■■ | US | Mixed | ■ | ? | ? | ■ | □ | □ | □ | □ |
| (11) | ■ | ? | ? | ■■■□ | US | Mixed | ■ | ? | ? | ■ | □ | □ | □ | □ |
| (12) | ■ | ISO 14040 | Not defined [e] | ■■□■ | AU [b2] | Primary | ■ | SimaPro | ReCiPe (H) | ■■ | ■ | ■ | □ | ■[g] |
| (13) | □ | ISO 14040 | 1000 trips | ■■□■ | US | Primary | □ | ? | TRACI | ■■ | ■ | □ | □ | □ |
| (14) | □ | ISO 14040 | 1 pallet | ■■□□ | US | Primary | ■ | SimaPro | ? | ■ | □ | □ | □ | □ |
| (15) | ■ | ? | ? | ■■■□ | US | Mixed | ■ | ? | ? | ■ | □ | □ | □ | □ |
| (16) | ■ | ISO 14040 | 1000 kg products delivered | ■■□■ | CZ | Primary | □ | GaBi | ReCiPe (H) | ■■ | ■ | ■ | ■ | ■ |

[a] – the system boundaries were denoted as follows: ■ - raw materials acquisition, ■ - pallet manufacturing, ■ - use phase impacts, ■ - end-of-life;

[b] – abbreviations according to ISO 3166 country codes, [b1] – RER is for Europe, [b2] – Australia and South East Asia;

[c] – transparent enough to replicate the key elements of the study;

[d] – one square is for global warming potential, while another is for other impact categories if any;

[e] – the function and functional units were not defined since the use phase of transportation was excluded which is the function of the pallet;

[f] – the production of raw materials such as steel nails, paint, electricity and energy is not considered;

[g] – deforestation was accounted for in the case of wooden pallets so that timber production for pallets would lead to deforestation in 70% of cases.

**Figure 3.** LCA-related criteria and data retrieved from reviewed articles. A filled square (■) denotes inclusion, while an empty one (□) indicates exclusion. A question mark (?) refers to a situation when a specific parameter was not mentioned or could not be identified from the reviewed study.

## 5. Pallets Studied

All in all, 43 pallets of different compositions, dimensions, structures, and management strategies were identified in the studies reviewed (Figure 4). Most studies focused on pooled pallets rather than on buy/sell or single-use pallets. The latter two were usually just referenced as alternative pallet management strategies. Stringer pallets were more commonly studied in the United States, where they represent 50%–54% of the total market share, whereas block pallets make up just 17%–20% [34]. In total, 15 of the cases studied involved GMA-sized pallets, which are the most commonly used pallets in the United States and account for 65% of the market share [35].

The pallets sized 1200 mm × 800 mm (i.e., EUR or European Pallet Association (EPAL) pallets), are the most commonly studied pallets in Europe; this is perhaps due to their abundant availability. The European Pallet Association (EPAL) [36] stated that in 2017 they produced 88.3 million EUR pallets. Regardless of the management strategy or the structure of the pallet, most of the pallets in the reviewed studies, were made of wood (*n* = 32) while the remaining cases studied plastic pallets (*n* = 8), cardboard (*n* = 2), and composite (*n* = 1).

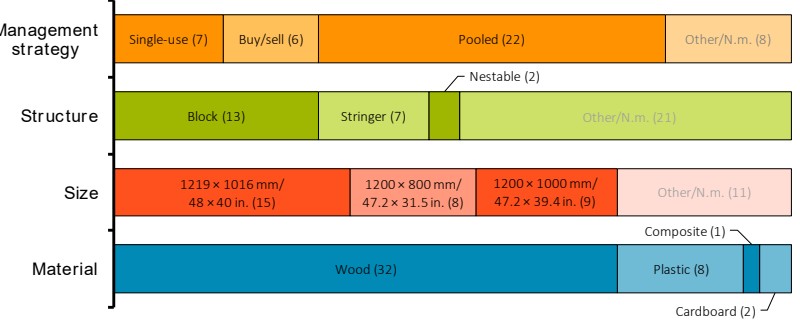

**Figure 4.** Types of the pallets studied organized by management strategy, structure, size, and material. N.m. – not mentioned.

The performance of the studied pallets expressed through their carrying capacity, the number of cycles, or the expected lifetime, ranged across the studies reviewed. The carrying capacity was seldom stated. The wooden pallet was given the capacity of 453–1350 kg depending on the management strategy [26,27], 1000 kg [19], or 1500 kg [30]. The plastic pallet was estimated to withstand higher loads of 1500 kg [30] or 1810 kg [20]. The wooden pallets were expected to last for 10 years in the majority of the studies analyzed. The largest variation in the studies reviewed was for the number of cycles, which is shown in Figure 5. The wooden pallets were mostly modeled to be used for 5 to 30 cycles, while occasionally performing at up to 90 cycles. The plastic pallets, however, have longer service lives which would last for 50–100 cycles. Shorter lives of five cycles and longer lives of 300 cycles were also considered. The longer service life of plastic pallets is due to its higher strength and better resistance to weathering. On the other hand, plastic pallets cannot be repaired, unlike wooden pallets, thus requiring better handling conditions to ensure a long life.

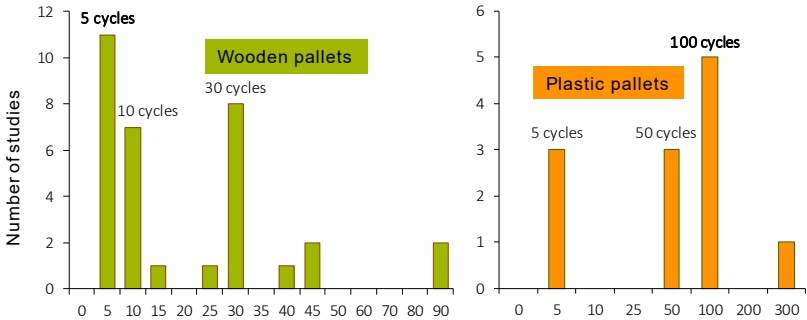

**Figure 5.** The number of cycles of wooden (left) and plastic (right) pallets.

## 6. Life Cycle Inventory

The studies which presented their inventory data for the production of pallets were analyzed further. Table 2 compiles the life cycle inventory data from the articles reviewed per single wooden or plastic pallet with specified dimensions and mass, if available. The presented data is reflective of the inputs during the production process in the form of intermediate flows.

**Table 2.** Life cycle inventory of the production of wooden and plastic pallets.

| No. | Pallet Size (mm) | Stated Pallet Mass (kg) | Inputs | | | | | | |
|---|---|---|---|---|---|---|---|---|---|
| | | | Wood (kg) | Plastic (kg) | Nails (kg) | Paint (kg) | Electricity (kWh) | Heat (MJ) | Fuel |
| (2)[a] | 1200 × 800 | 24.5 | 25.2 | - | 0.430 | 0.042 | 0.70 | 6.1 | 0.017 L, LFO [o] |
| | 1200 × 800 | 24.3 | 25.5 | - | 0.430 | - | 0.70 | 5.7 | 0.017 L, LFO |
| (3) | 1219 × 1016 | - | 22.5 | - | 0.360 [b] | - | 0.95 | - | 0.18 L, diesel |
| | 1219 × 1016 | 20.4 | - | 20.5 | - | - | 8.5 | - | - |
| (5) | 1200 × 800 | 8.55 | 8.37 | - | 0.180 | - | - [c] | - | - |
| (6) | 1219 × 1016 | - | 13.8 | - | 0.324 [b] | - | 0.12 | 10.0 | - |
| | 1219 × 1016 | - | 29.7 | - | 0.378 [b] | - | 0.093 | 10.0 | - |
| | 1219 × 1016 | - | 31.6 | - | 0.459 [b] | - | 0.062 | 10.0 | - |
| (7) | 1200 × 800 | - | 17.1 | - | 0.272 | - | 0.20 | 0.60 | - |
| | 1200 × 800 | - | 11.9 [d] | - | 0.272 | - | 0.20 | - | - |
| (8)[e] | 1200 × 800 | - | 9.10 | - | 0.490 | - | 2.2 | - | 0.16 L, gas oil |
| (9) | 1165 × 1165 | - | 20.9–31.3 [f] | - | 0.390 | 0.14–0.15 | 0.47–1.00 | 0.39–1.56 | - |
| | 1200 × 1000 | - | 14.2–18.5 [f] | - | 0.530 | 0.083 | 0.50–1.00 | 1.04–1.56 | - |
| | 1165 × 1165 | 34.0 | - | 35.7 [g] | - | 0.36 [h] | 35.7 | - | - |
| | 1200 × 1000 | 19.4 | - | 19.4 | - | 0.20 [i] | 25.0 | - | - |
| (12) | 1200 × 1000 | 13.8 | - | 16.2 [j] | - | - | 190 | - | 0.0040–0.0060 L, diesel |
| | 1090 × 1090 | 15.5 | - | 18.3 [j] | - | - | 170 | - | 0.0036–0.0053 L, diesel |
| | 1200 × 1000 | 35.0 | - | 41.3 [j] | - | - | 430 | - | 0.0091–0.0136 L, diesel |
| | - | - | 13.0 | 13.0 | - | 0.310 | - | 0.29–0.58 | 0.60–0.90 |
| | - | - | 17.0 | 17.0 | - | 0.290 | - | 0.27–0.55 | 0.57–0.81 |
| | - | - | 37.5 | 35.0–40.0 | - | 0.690 | - | 0.61–1.40 | 0.85–1.45 |
| | - | - | 6.50 | - | 6.5 [k] | - | 0.070 [h] | 8.38 | - | - |
| | - | - | 34.0 | - | 34.4 [k] | - | 0.39 [h] | 48.3 | - | - |
| (16) | 1200 × 800 [m] | - | 20.0–25.0 | - | - [n] | - | 0.10–0.13 | - | 0.032–0.039 L, diesel |
| | 1200 × 800 | 25.0 | - | 25 [l] | - | - | 14.1 | - | 0.63 L, diesel |

[a] The data was converted from the functional unit stated in the paper; [b] calculated using the weight of a single nail of 4.5 g; [c] the amount of electricity, heat, and fuels used was not directly specified in the paper and could not be retrieved from the information presented in the study with a high degree of certainty; [d] technical wood was used in the manufacture; [e] calculated based on the data provided in the study cited; [f] calculated from the volumetric data using the density of wood of 474 kg/m³; [g] the pallet was made of recycled high density polyethylene; [h] carbon black was used in the manufacture as a UV inhibitor; [i] color pigments were used in the manufacture; [j] the pallet was made of recycled plastic; [k] the pallet contained 85% virgin HDPE and 15% recycled HDPE; [l] the palled contained 70% virgin plastic, 20% industrial waste plastic, and 10% recycled plastic pallets; [m] the data for the wooden pallet were deduced from annual data using the mass of cuttings as a reference flow and assuming the mass of a single pallet of 20–25 kg; [n] the mass of nails deduced from the yearly data was significantly lower than the average data from other literature, so the value was omitted, [o] light fuel oil.

### 6.1. Wooden Pallets

The key inputs to the production of wooden pallets are wood itself, such as timber or particleboard, nails to fasten the wood, electricity for the process equipment, and thermal energy or fuel for phytosanitary treatment [37,38]. Additionally, wood can be pained, which is a common practice for pooled pallets to enable their tracking and distinguish them from the pallets of other companies. Finally, phytosanitary treatment can be implemented through fumigation of the pallets with methyl bromide, radiofrequency heating, or conventional heat treating. The latter being the most common method of treatment [20]. The International Phytosanitary Measure (IPSM) 15 mandates that wooden packaging must be heat-treated or fumigated with methyl bromide, stamped with the appropriate labeling, and be de-barked in order to cross international borders. This is to mitigate pests and bugs from spreading via trade.

The mass of the required materials varied significantly across the studies. This can be partly attributed to the differing dimensions of the reviewed pallets. Also, the pallet management strategy affects the requirements set on the pallets: the single-use pallets are not intended to last, so thinner wood is used, while the opposite holds true for the polled pallets. The weight of the wood required for a wooden pallet ranged from 8.4–40 kg with an average value of 21.4 ± 8.8 kg. Considering the EUR pallets only (1200 mm × 800 mm), the weight of the wood required ranged from 8.4–25.0 kg with the average value of 17.1 ± 6.9 kg. However, the average weight calculated based on the studies reviewed seems to be underestimated if compared to the bottom-up estimation of the weight of the pallet based on the requirements of the EUR pallet [37]. Table 3 calculates the weight of a wooden EUR pallet based on the number of components and their dimensions from the SFS-EN 13698-1 standard. The total weight of the pallet using a bottom-up approach is 21.82 kg. According to the European Pallet Association [39], the approximate weight of a EUR pallet is 25 kg. The exact weight of a pallet depends also on the density of wood used, as well as its moisture content.

**Table 3.** Dimensions, number of components, and weight of a wooden European (EUR) pallet.

| Component | Number of Components | Length (mm) | Width (mm) | Thickness (mm) | Volume (m$^3$) | Density (kg/m$^3$) | Weight (kg) |
|---|---|---|---|---|---|---|---|
| Bottom deck lead board | 2 | 1200 | 100 | 22 | 0.0053 | | 2.50 |
| Top deck lead board | 2 | 1200 | 145 | 22 | 0.0077 | | 3.63 |
| Central bottom deck board | 1 | 1200 | 145 | 22 | 0.0038 | | 1.81 |
| Stringer board | 3 | 800 | 145 | 22 | 0.0077 | 474 | 3.63 |
| Central top deck board | 1 | 1200 | 145 | 22 | 0.0038 | | 1.81 |
| Intermediate top deck board | 2 | 1200 | 100 | 22 | 0.0053 | | 2.50 |
| Outer skid block | 6 | 145 | 100 | 78 | 0.0068 | | 3.22 |
| Center skid block | 3 | 145 | 145 | 78 | 0.0049 | | 2.33 |
| Nails | 78 | - | - | - | - | - | 0.38 |
| Total | | | | | | | 21.82 |

The weight of the nails required ranged from 0.18–0.69 kg per wooden pallet with an average of 0.38 ± 0.12 kg. Considering only EUR pallets, the weight of the nails required varied from 0.18 to 0.49 with the average weight of 0.35 ± 0.11 kg. Given that 78 nails are required to produce a EUR pallet according to the standard, the weight of a single nail would be 4.5 g.

The electricity demand for the production of an EUR pallet ranged significantly from 0.12 to 2.2 kWh per pallet with the average of 0.69 ± 0.73 kWh. The value is significantly lower than that of a plastic pallet, as discussed later. Apart from electricity, heat is used in the production of wooden pallets, yet its quantification is challenging because thermal energy required is given as heat in some studies and as fuel in others. In those studies which give thermal energy demand directly, the average value was 4.1 ± 2.5 MJ per EUR pallet. The diesel or light fuel oil demand ranged from 0.017–0.16 L per pallet.

*6.2. Plastic Pallets*

The production of plastic pallet requires plastic itself, which can be either virgin plastic or recycled, electricity for thermoforming of plastic, and diesel for the machinery operating at the plant. No phytosanitary treatment is required for plastic pallets. The mass of plastic pallets equals the mass of plastic required, as well as possible plastic waste generation. Due to lack of standards on plastic pallets, their dimensions vary significantly, though they often conform to the dimensions of the wooden pallets. The shares of the studied plastic pallets are shown in Figure 6.

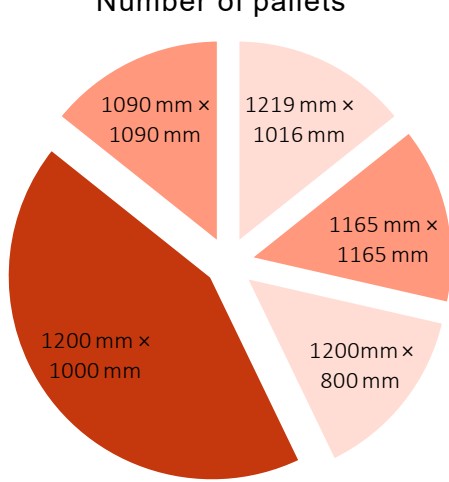

**Figure 6.** The number of plastic pallets studied by their size.

Since the plastic pallets are molded, the inputs of the plastic were normalized to 1 m$^2$ of the pallet surface to ensure cross-comparability of the studied pallets. The minimum mass of plastic required per 1 m$^2$ of the plastic pallet was 13 kg, whereas the maximum weight was 34 kg. The average mass of the plastic needed was 21 ± 7 kg/m$^2$, which equals 20 kg per EUR pallet, 25 kg per FIN pallet (1200 mm × 1000 mm), or 26 kg per GMA pallet.

Unlike wooden pallets, the production of plastic pallets requires more electricity which is needed to melt the plastic and inject it into the mold. The minimum electricity demand is 6.8 kWh and the maximum is 359 kWh per 1 m$^2$ of the pallet surface. The average electricity demand is 104 ± 120 kWh/m$^2$. To compare, the electricity demand for a wooden EUR pallet is 0.69 ± 0.73 kWh. Therefore, the carbon intensity and the type of electricity required has a significantly more pronounced impact on plastic pallet production. Small diesel consumption was also included in some studies at the level of 0.04–0.014 L [28] and 0.63 L [30].

## 7. Life Cycle Impact Assessment

The reviewed studies mainly focused on the impacts on climate change (Figure 3), whereas some of the studies also assessed other impact categories. For this reason, the focus on this section is on the systematic assessment of the results of the carbon footprint while also giving an overview of other impact categories based on a limited number of studies.

### 7.1. Carbon Footprint

The carbon footprint of a pallet, also referred to as global warming potential or climate change, is a sum of greenhouse gas emissions and removals occurring during a pallet's life cycle and expressed as a mass of carbon dioxide equivalents (e.g., kg $CO_2$-eq.). The carbon footprint can be calculated using specific guidelines of the SFS-EN ISO 14067 standard [32], which solely focuses on climate change impacts, or following the generic requirements of SFS-EN ISO 14040 [31] standard by using only one impact category of climate change. Other standards also exist, yet they were not widely followed in the studies reviewed. Caution should be given to possible differences in the accounting of greenhouse gases by different impact assessment methods used in the studies reviewed (i.e., IPCC, CML2000, TRACI, and ReCiPe (H)).

Table 4 lists the carbon footprints from the studies reviewed per life cycle stages recommended by the methodology for product environmental footprint calculations [40]. The results are given as total carbon footprint per one pallet and/or per one trip depending on the data available in the studies reviewed for the conversion of the results. Please note that not all data presented in the original research studies exactly matched the classification of life cycle stages used in this study. Therefore,

the values presented per life cycle stage should be read with caution and if the needed, the original article should be referred to.

**Table 4.** Global warming potentials of wooden and plastic pallets. The total might not equal the sum of columns due to rounding.

| No. | Material | Pallet Size (mm) | GWP Per Pallet (kg CO$_2$-eq.) | GWP Per Cycle (kg CO$_2$-eq.) | Life Cycle Stage [1] | | | | |
|---|---|---|---|---|---|---|---|---|---|
| | | | | | RMA and PP | MAN | DIS | USE | EOL |
| (1) | Wood [2] | 1219 × 1016 | | 1.9 | | 0.3 | | 1.6 | |
| | Wood [2] | 1219 × 1016 | | 2.5 | | 1.1 | | 1.4 | |
| | Wood [2,3] | 1219 × 1016 | | 4.4 | | 2.9 | | 1.5 | 0.064 |
| (2) | Wood | 1200 × 800 | 8.2 | | −39 | 0.91 | 57 | −12 | 2.1 |
| | Wood | 1200 × 800 | −26 | | −40 | 0.88 | 10 | −0.75 | 4.0 |
| (3) | Wood | 1219 × 1016 | 21 | 1.2 | | 10 | | 9 | 2.0 |
| | Plastic | 1219 × 1016 | 166 | 1.3 | | 50 | | 110 | 5.8 |
| (4) | Wood | | 17 | 0.67 | | | | | |
| (5) | Wood [2] | 1200 × 800 | 2.3 | | 3.1 | 0.18 | | | −1.0 0.19 [5] |
| (6) | Wood [3] | 1219 × 1016 | −5.6–2.1 | | 1.7 | 0.23 | | | 0.11 −7.6 0.42 [5] |
| | Wood | 1219 × 1016 | −13–6.5 | | 3.0 | 0.21 | | 3.0 [4] | 0.27 −19 0.44 [5] |
| | Wood | 1219 × 1016 | −11–7.6 | | 3.5 | 0.19 | | 3.7 [4] | 0.22 −18 |
| (7) | Wood | 1200 × 800 | 4.0 | | 3.8 | 0.21 | | | |
| | Composite | 1200 × 800 | 3.5 | | 3.4 | 0.12 | | | |
| (8) | Wood [2] | 1200 × 800 | 9.9 c | | | | | | |
| | Wood | 1200 × 800 | 8.1 | | 6.2 | 1.3 | 0.57 | | |
| (9) | Wood | 1200 × 1000 | 20–26 [6] | 0.44–0.58 | | | | | |
| | Wood | 1165 × 1165 | 50–61 [6] | 0.60–0.73 | | | | | |
| | Plastic | 1200 × 1000 | 61 | 0.98 | | | | | |
| | | 1165 × 1165 | 102 | 1.6 | | | | | |
| | Wood [2] | 1200 × 1000 | 3.1 | 1.6 | | | | | |
| | | 1165 × 1165 | 2.2 | 1.1 | | | | | |
| (11) | Wood | 1219 × 1016 | 3.7 | | | | | 3.7 [7] | |
| (12) | Plastic | 1200 × 1000 | 3.7 | 1.4–4.1 | | 3.7 | | | |
| | Plastic | 1090 × 1090 | 4.1 | 2.0–6.1 | | 4.1 | | | |
| | Wood | 1200 × 1000 | 8.8 | 7.2–22 | | 8.8 | | | |
| | Plastic | 1200 × 1000 | 22 | 9.2–28 | | 22 | | | |
| (14) | Wood | 1219 × 1016 | −0.20 | | | | | | −0.20 [8] |

[1] The life cycle stages were abbreviated as follows: RMA and PP (raw materials acquisition and pre-processing), MAN (manufacturing stage), DIS (distribution stage), USE (use phase), and EOL (end-of-life); [2] the data were retrieved from graphs; [3] the type of the pallet is a single-use pallet; [4] the value is the average of the worst and best handling conditions; [5] the values are for the following end-of-life options: landfilling, mulching, and incineration with energy recovery; [6] the smaller value is for the softwood pallet and the bigger one is for the hardwood pallet; [7] the impact from repair of a pallet throughout its lifetime; [8] the impact of a single repair. Avoided impact originates from avoided lumber production and recycling of steel scrap which together outweighs the impact of electricity and nails provision.

The results show a large variation in the climate change impacts of wooden pallets. The impact of a wooden EUR pallet ranged from −26 to 9.9 kg CO$_2$-eq. per pallet. The negative impact on climate change was due to carbon sequestration of wood (−40 kg CO$_2$-eq. per pallet), which was accounted for during wood harvest in the study by Gasol et al. [19], yet the end-of-life was mostly recycling which was chipping of wood, proposing wooden pallets as carbon stock. The results by Gasol et al. [19], however, differed for the high reuse intensity pallet, which had the total carbon footprint of 8.2 kg CO$_2$-eq., which was due to higher impacts from transportation needed to ensure forward and reverse logistics of the pallets. The carbon footprint of the EUR pallet in other studies was on average 6.1 ± 3.1 kg CO$_2$-eq.,

where the majority of the impact originated from the acquisition of the raw materials. However, if the pallets were designed to be reused, the impact from transportation is expected to be higher, as shown in the studies by Gasol et al. [19] and Kurisunkal [20]. The study by Carrano et al. [23] evaluated the environmental impacts of different end-of-life scenarios for the pallets, land filling, mulching, and inclination with energy recovery. The authors showed that incineration with energy recovery and substitution of electricity results in emissions savings at the level of $-7.6$ to $-19$ kg $CO_2$-eq. per pallet. This is possible because of the biogenic nature of wood, whose incineration does not affect climate change as opposed to other fuels, which are being replaced with wood.

The climate change impacts from the product systems studying plastic pallets were in general higher compared to wooden pallets. Kurisunkal [20] estimated the impact to be 166 kg $CO_2$-eq. per pallet, if the pallet makes 100 trips. Bengtsson and Logie [25] reported the impact at the level of 61–102 kg $CO_2$-eq. depending on the country of pallet production and its size. The researchers at Edge Environment Pty Ltd. [28] assessed the impact of pallets made of recycled plastic to be 3.7–4.1 kg $CO_2$-eq., whereas that of a conventional plastic pallet was 22 kg $CO_2$-eq. The difference between waste and virgin plastic is in the zero-burden approach used for waste plastic. Furthermore, plastic pallets in the study of Edge Environment Pty Ltd. [28] were modeled as being landfilled, which has only a small impact on climate change of 0.088 kg $CO_2$-eq. per kg of material or 2.2–2.7 kg $CO_2$-eq per pallet. On the contrary, if the pallets were incinerated, the impact would increase to approximately 50 to 60 kg $CO_2$-eq. per plastic pallet.

### 7.2. Other Impact Categories by Studies

Apart from carbon footprints, some studies evaluated the acidification potential, eutrophication potential, ozone layer depletion potential, toxic impact, use of water, and land occupation, among other impacts. The impact assessment methods varied across the studies: CML 2000 was used by Gasol et al. [19], Impacts 2002+ by Kurisunkal [20], ReCiPe 2008 by Niero et al. [22], ReCiPe (v1.10) by Bengtsson and Logie [25], ReCiPe (v1.12) by Edge Environment Pty Ltd [28], and ReCiPe 2016 (v1.1) by Kočí [30]. Owing to the variation in the accounting of impacts in different methods and variation of characterization factors within the same method but different versions, the studies were not cross-compared. Instead, alternative pallets or their management strategies were discussed.

#### 7.2.1. Reuse Intensity

Gasol et al. [19] found that the pallets with the higher reuse intensity (30 cycles) performed better than the pallets with the lower reuse intensity (4.4 cycles) when considering their environmental impacts per 1000 cycles. However, the impacts per pallet were higher in the case of the pallets indented to be reused frequently. They are built to be more durable and therefore consume more resources. The energy consumed in the studied system was 62 MJ for the high reuse system and 171 MJ for the low reuse system. The impact per pallet was 1.9 and 0.75 MJ, respectively. A similar trend was seen for the other impact categories studied.

#### 7.2.2. GMA-Sized Wooden and Plastic Pallets

The study by Kurisunkal [20] identified a significant difference in the impacts of a wooden block-type GMA pallet and a plastic pallet of the same size. The only impact category where plastic pallets showed better performance was land occupation scoring 0.21 m$^2$ for a plastic pallet and 2.6 m$^2$ for a wooden pallet. For the other impact categories, the plastic pallet showed higher impact by 1.5–148,499 times. The largest difference was for carcinogenic impacts which were 0.59 kg $C_2H_3Cl$-eq. for a wooden pallet and 87,169 kg $C_2H_3Cl$-eq. for a plastic pallet, with 99% of the impact coming from the supply of plastic.

### 7.2.3. Pallets in Australia and China

According to Bengtsson and Logie [25], the production of a wooden (softwood) 1165 mm × 1165 mm pallet in Australia has an endpoint impact of 6.8 Pt versus 3.1 Pt originating from the production of a 1200 mm × 1000 mm wooden (softwood) pallet in China. Also, the consumption of fossil fuels is higher in Australia than in China being 20 kg oil eq. and 6.0 kg oil eq., respectively. Even if Australian pallets performed better (12 pallets is needed for 1000 trips) than the Chinese ones (22 pallets is needed), the order of the results remained the same. On the contrary, plastic pallets production in Australia has lower impacts than in China: 11 Pt vs. 13 Pt and 38 kg oil eq. vs. 54 kg oil eq., respectively. The same applies to the functional unit of 1000 trips since the pallets are expected to have the same number of uses during their lifetimes.

### 7.2.4. Pallets Made of Waste Plastic or Tropical Wood

The study conducted by Edge Environment Pty Ltd [28] and commissioned by Range International, which is a company manufacturing plastic products from waste plastic, analyzed the impacts of producing and utilizing pallets from waste plastic using a zero-burden approach and compared the results to a wooden pallet made of unsustainably sourced tropical wood, and a conventional plastic pallet. The results indicated that the waste plastic pallets were 3–801 times better than the conventional plastic pallet depending on the impact category analyzing the results per trip. The largest difference was recorded for the damage on ecosystems which was $1.6 \times 10^{-5}$ species a year per trip for the conventional plastic pallet and $2 \times 10^{-8}$ to $3.5 \times 10^{-8}$ per trip for the waste plastic pallets. The freshwater eutrophication potential, terrestrial acidification potential, and fossil fuel depletion potential had a smaller difference of 3–4 times. The waste plastic pallets also performed better, environmentally, than the wooden pallets, yet the difference was smaller, 3–13 times. The largest difference occurred for the freshwater eutrophication potential and fossil fuel depletion potential. The assumption of landfilling as the waste plastic pallet's end-of-life scenario was largely responsible for the lower impact.

## 8. Conclusions

The environmental impact associated with the production, use, and disposal of various pallets has been assessed in several studies identified and reviewed in this paper. In total, 16 studies were identified. The reviewed studies employed different approaches to life cycle assessment. This paper systematically analyzed those studies and tabulated key methodological assumptions made with the inventory data available and analyzed their results.

The most studied pallets were block-type 1219 mm × 1016 mm ($n$ = 13/43)/48 in. × 40 in. ($n$ = 15/43), pallets made of wood ($n$ = 32/43), and intended to be pooled ($n$ = 22/43). The pallet market in the United States was mostly studied, while some studies in the context of European and Asian countries were identified. As the function of a pallet is to transport cargo, the recommended functional unit is the number of trips through the supply chain that a pallet can make in its lifetime. This allows the difference between types of pallets to be accounted for. Also, the load can be considered in the functional unit, yet a high uncertainty on the actual carrying capacity should be addressed. Therefore, the load-based functional unit can be utilized in LCA studies where pallets are only a part of the studied product system to be able to normalize the impact per product studied.

There was a significant amount of inventory data on the production of wooden and plastic pallets, while data on pallets made of wood–polymer composites was missing. A large variation in the number of raw materials required to produce the pallets was observed. Wood consumption ranged from 8.4–40 kg with an average of 21.4 ± 8.8 kg per pallet, considering pallets of all sizes. Wood consumption for the production of a EUR pallet ranged from 8.4–25 kg with the average value of 17.1 ± 6.9 kg. Considering that the average weight of a EUR pallet given in the literature is 20–25 kg, the average value identified during the review is expectedly underestimating the mass of wood required. Nail consumption for a EUR pallet ranged from 0.18 to 0.49 with the average weight of

$0.35 \pm 0.11$ kg, giving 4.5 g for the average weight of a nail because 78 nails are required per EUR pallet. Electricity consumption showed the highest variation between 0.12 and 2.2 kWh per EUR pallet with an average of $0.69 \pm 0.73$ kWh.

Unlike wooden pallets, there was lower variation in the inventory data of plastic pallets, which can be related to a fewer number of studies focusing on plastic pallets. Since plastic pallets are produced by injection molding, their weight is more uniform than for the wooden pallets. For this reason, the inventory for plastic pallets is given per square meter of the pallets. Plastic consumption ranged from 13–34 kg with an average weight of $21 \pm 7$ kg per 1 $m^2$. Electricity consumption was significantly higher compared to the wooden pallets and ranged from 6.8–359 kWh per 1 $m^2$ with the average value of $104 \pm 120$ kWh per 1 $m^2$.

Global warming potential was the most commonly studied impact category. The results for a EUR pallet ranged from −26 to 9.9 kg $CO_2$-eq. per pallet. The negative value is due to carbon sequestration of wood which was accounted for during wood harvest. At the same time, the wood was chipped at its end-of-life, thus eliminating the release of biogenic carbon back to the atmosphere (i.e., partial carbon footprint). Acquisition of raw materials and transportation of pallets during their use are found to have the largest impact during their lifecycles.

Plastic pallets were found to exert a higher impact on climate change compared to wooden pallets. The impact was calculated to range from 22–166 kg $CO_2$-eq. per pallet if virgin plastic is used and 3.7–4.1 kg $CO_2$-eq. per pallet, if waste plastic is used. The use of waste plastic reduced the impacts due to the zero-burden approach. Also, landfilling of plastic pallets does not have a large impact on climate change, thus giving plastic a better result for climate change. However, plastic pallets were found to generally have higher impacts across other impact categories, such as carcinogenic impacts, fossil fuel depletion, acidification, and eutrophication.

Based on the literature reviewed, it can be recommended that future studies include a standardized minimum amount of information on pallets, data used, and impact assessment. The pallets should be characterized based on their structure (stringer-type or block-type), dimensions, materials used, and finally the management strategy (single-use, buy/sell, or pooled). Also, information on the pallet load and the number of trips during a lifecycle should be stated to ensure that the data can be converted either to a trip or a pallet, which are the common functional units. The inventory used should be clearly presented to ensure comparability of the data with other literature. The pallet's end-of-life scenario should be specified since it has a high impact on the results. Finally, the results of the studies should be clearly presented per life cycle stages to ensure transparency and comparability of the results.

**Author Contributions:** Conceptualization, I.D., M.K., E.E., and M.H.; methodology, I.D. and M.H.; data analysis, I.D. and M.K.; writing—original draft preparation, I.D.; writing—review and editing, E.E., M.K., and M.H.; visualization, I.D.; supervision, M.H.; funding acquisition, M.H.

**Funding:** This research was funded by the Life IP on waste—Towards a circular economy in Finland (LIFE-IP CIRCWASTE-FINLAND) project (LIFE 15 IPE FI 004). Funding for the project was received from the EU Life Integrated program, companies, and cities. The APC was funded by the Department of Sustainability Science, LUT Universit.

**Conflicts of Interest:** The authors declare no conflict of interest.

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
