# Peer review of "Wooden and Plastic Pallets: A Review of Life Cycle Assessment (LCA) Studies"

_sustainability, doi:10.3390/su11205750_

Round 1
Reviewer 1 Report
Dear Authors,
Your paper is an accurate description of the subject.
Two typos:
- line 50 FUN size 1200 x 1000 (instead of 800)
- line 175 CLM 2000 (insteda of 200)
Author Response
Response to Reviewer 1 Comments
Honored Reviewer 1, we would like to thank you for the provided feedback on the paper. The points raised were responded as follows.
Point 1: line 50 FUN size 1200 x 1000 (instead of 800)
Response 1. The size of the pallet was changed to the correct one as pointed out by the Reviewer. Now it reads: "...FIN size – 1200 mm x 1000 mm..."
Point 2: line 175 CLM 2000 (insteda of 200)
Response 2. The authors corrected the name of the characterization method to CML 2000 as pointed out by the reviewer.
Reviewer 2 Report
A review of the manuscript entitled “Wooden and plastic pallets: a review of life cycle assessment (LCA) studies”
The manuscript is written logically and clearly.
The main suggestion
Lines 396-399. The authors propose using the following definition of the functional unit "the number of trips through the supply chain that a pallet can make in its lifetime". However, the authors also pay attention to the actual carrying capacity of a pallet. Perhaps the authors should recommend the following functional unit “the number of trips through the supply chain that a pallet can make in its lifetime normalized to the actual pallet carrying capacity.”
The minor suggestion
Line 397. It seems that the manuscript contains misprint: “though” instead “through”. Please check it through all the text.
Table 4. The total number of kilograms of the pallet must have two digits after the dot.
Author Response
Response to Reviewer 2 Comments
Honored Reviewer 1, we would like to thank you for the provided feedback on the paper. The points raised were responded as follows.
Point 1: Lines 396-399. The authors propose using the following definition of the functional unit "the number of trips through the supply chain that a pallet can make in its lifetime". However, the authors also pay attention to the actual carrying capacity of a pallet. Perhaps the authors should recommend the following functional unit “the number of trips through the supply chain that a pallet can make in its lifetime normalized to the actual pallet carrying capacity.”
Response 1. We are thankful for this important comment! Indeed, the mass of cargo transported is an important parameter, which can be used when comparing pallets of different structures which can have different loading capacities.However, the transportation of the pallets during their actual use by the customers have usually been excluded from LCAs of pallets. For this reason the FU was stated as the number of trips regardless of the mass of the transported cargo. However, the proposed functional unit accounting for the mass of transported cargo is of value for LCA studies where pallets are a part of the products system used in logistics of studied products. A corresponding clarification was added as follows on LL 398-401: "Also, the load can be considered in the functional unit, yet a high uncertainty on the actual carrying capacity should be addressed. Therefore, the load-based functional unit can be utilized in LCA studies where pallets are only a part of the studied product system to be able to normalize the impact per product studied."
Point 2: Line 397. It seems that the manuscript contains misprint: “though” instead “through”. Please check it through all the text.
Response 2: Both the word "though" and the word "through" were checked in the manuscript. Two changes were made where "through" was misspelled with "though". The authors apologize for the grammatical mistake. The article has been initially proofread by a native English speaker.
Point 3: Table 4. The total number of kilograms of the pallet must have two digits after the dot.
Response 3: The authors added the second decimal in the total mass of the pallet as requested. The value has also been adjusted in the text of the manuscript.
Reviewer 3 Report
The review of LCA of wodden and plastic pallets is well organised and scientific sound. The authors have performed a well organised literature research on pallets. I have just minor comments and address all chapters in the review:
The introduction and abstract are well done.
Chapter 2: the method (here: review) is well presented with figure 1
Chapter 3: good overview on the pallet classification and the studies in the table 1 and figure 2.
Chapter 4:
I am missing the link between number of trips (line 139) and the km driven. I assume that there is a dependency between the distance driven and the usage of pooled pallets. The shorter the trip , the higher the chance to use a pooled pallet? And the opposite with longer distances: long distance - higher useage of single-use pallets? This also impacts the number of cycles?
Table 2 is a very good overview. I was surprised that there are so many studies that did not even mention the used LCA software. Please check if the studies mention the database used. I assume that most of them use ecoinvent. But maybe not? As the table is already very informative this would be the only fact that is missing.
Chapter 5:
Figure 4 is a bit misleading. What does the different coloring of the bars mean? As there is no legend for it , it remains unexplained.
What means the y-achsis : frequency? Frequency is misleading as the other achsis means the number of cycles. Please explain in the text or use another description.
Chapter 6 and 7 are well done, whereas table 5 is very good as you mention the single life stages.
Finally, I have a last comment for chapter 8. Landfilling of plastic is forbidden in Europe, therefore it cannot be an option, as stated in line 426. Maybe, it is an option on other continents. But according to the waste hierarchy , landfilling should always be the very last solution.
Author Response
Response to Reviewer 3 Comments
Honored Reviewer 3, we would like to thank you for your extensive and profound feedback on the paper. The points which needed attention were responded as follows.
Point 1: I am missing the link between number of trips (line 139) and the km driven. I assume that there is a dependency between the distance driven and the usage of pooled pallets. The shorter the trip , the higher the chance to use a pooled pallet? And the opposite with longer distances: long distance - higher useage of single-use pallets? This also impacts the number of cycles?
Response 1. While there must be an economically justified reason for choosing either pooled or single-use (or one way) pallets depending on the distance between the supplier and the customer, the choice of the pallet management strategy has not been a subject of this review paper. Instead, we tried to distinguish between the three management strategies and to show how those were applied in the studied reviewed. If pallets are pooled then the number of used uses is high, which is accounted for in LCA to make it comparable with single-use or buy/sell pallets.
Point 2: Table 2 is a very good overview. I was surprised that there are so many studies that did not even mention the used LCA software. Please check if the studies mention the database used. I assume that most of them use ecoinvent. But maybe not? As the table is already very informative this would be the only fact that is missing.
Response 2. The authors are thankful for this valuable comment. This was out initial idea, too, to complement the table with the source of inventory data used, but it soon turned out to be impossible since there are various sources of inventory used in the studies ranging from mostly secondary data to the data from companies and a mixture of those. Furthermore, identification of a single source of data which prevailed the studies might be biasing because it is not the number of datasets used, but their significant for the results which matter. We apologize for not being able to add the missing column to Table 2.
Point 3: Figure 4 is a bit misleading. What does the different coloring of the bars mean? As there is no legend for it , it remains unexplained.
Response 3. The authors apologize for not clealy specifying the difference in shades in te data series presented. Initially, the more denseshades were used for the cases with the largest frequency to emphasize the key results. However, the authors decided to keep only one color without shading to avoid confusion. The results remained unchanged.
Point 4: What means the y-achsis : frequency? Frequency is misleading as the other achsis means the number of cycles. Please explain in the text or use another description.
Response 4. In Figure 4, the frequency refers to the number of cases stating a specific number of cycles for the pallets during their lifecycle. The term frequency has now been replaced with "Number of studies" to avoid confusion.
Point 5: Finally, I have a last comment for chapter 8. Landfilling of plastic is forbidden in Europe, therefore it cannot be an option, as stated in line 426. Maybe, it is an option on other continents. But according to the waste hierarchy , landfilling should always be the very last solution.
Response 5. The authors are thankful for raising so important aspect. However, according to our best knowledge, there is no EU-wide ban on landfilling of plastic, yet some attempts exist. E.g. EU plans to partly ban single-use plastic packaging, but this does not concern other products made of plastic. Aslo, some countries like Finland introduced bans on landfilling of organic waste (http://www.cewep.eu/wp-content/uploads/2017/12/Landfill-taxes-and-bans-overview.pdf) but those cases are still seldom. Finally, there are recycling targets for MSW, plastic packaging, and C&D waste, but those dont directly relate to the ban of plastic, but more to their efficient use.